# PriorCCI: Interpretable Deep Learning Framework for Identifying Key Ligand–Receptor Interactions Between Specific Cell Types from Single-Cell Transcriptomes

**DOI:** 10.3390/ijms26157110

**Published:** 2025-07-23

**Authors:** Hanbyeol Kim, Eunyoung Choi, Yujeong Shim, Joonha Kwon

**Affiliations:** 1Bioinformatics Branch, National Cancer Center, Goyang 10408, Republic of Korea; googija92@ncc.re.kr (H.K.); choieunyoung5959@ncc.re.kr (E.C.); shimyj@ncc.re.kr (Y.S.); 2Department of Public Health & AI, Graduate School of Cancer Science and Policy, National Cancer Center, Goyang 10408, Republic of Korea

**Keywords:** cell-cell interaction, convolutional neural network, explainable AI, prioritization, single-cell RNA-seq data, tumor microenvironment, endothelial cell, angiogenesis

## Abstract

Understanding the interactions between specific cell types within tissue environments is essential for elucidating key biological processes, such as immune responses, cancer progression, inflammation, and development, in both physiological and pathological studies. The predominant methods for analyzing cell–cell interactions (CCI) rely primarily on statistical inference using mapping or network-based techniques. However, these approaches often struggle to prioritize meaningful interactions owing to the high sparsity and heterogeneity inherent in single-cell RNA sequencing (scRNA-seq) data, where small but biologically important differences can be easily overlooked. To overcome these limitations, we developed PriorCCI, a deep-learning framework that leverages a convolutional neural network (CNN) alongside Grad-CAM++, an explainable artificial intelligence algorithm. This study aims to provide a scalable, interpretable, and biologically meaningful framework for systematically identifying and prioritizing key ligand–receptor interactions between defined cell-type pairs from single-cell RNA-seq data, particularly in complex environments such as tumors. PriorCCI effectively prioritizes interactions between cancer and other cell types within the tumor microenvironment and accurately identifies biologically significant interactions related to angiogenesis. By providing a visual interpretation of gene-pair contributions, our approach enables robust inference of gene–gene interactions across distinct cell types from scRNA-seq data.

## 1. Introduction

Single-cell RNA sequencing (scRNA-seq) has revolutionized the study of cellular interactions by enabling high-resolution analysis at the cellular level. It allows researchers to examine co-expression patterns between cell types and gain insights into the functional roles of specific gene–gene interactions [1,2,3]. By comparing expression profiles across defined cell type groups, scRNA-seq data revealed how cells communicate and coordinate functions within tissue environments, contributing to our understanding of tissue organization and disease mechanisms [4,5].

A wide range of computational tools has been developed to infer and analyze cell–cell communication using single-cell transcriptomic data [6]. These tools adopt diverse algorithms, statistical frameworks, and assumptions to reconstruct intercellular interactions, predict signaling pathways, and identify key mediators of communication [7]. Many studies utilize ligand–receptor expression patterns to infer potential interactions, offering complementary perspectives that, when integrated, provide a more comprehensive understanding of signaling networks in complex biological systems [3,8].

Among these tools, CellChat is a well-established platform that leverages ligand–receptor interaction databases and scRNA-seq data to infer signaling activities and estimate both the strength and directionality of communication pathways [7,9,10]. CellPhoneDB, another widely used resource, combines a curated ligand–receptor database with statistical testing to predict significant associations between cell types [11]. NicheNet takes a receptor-centric approach, incorporating regulatory networks of receptor-mediated pathways and transcription factors to predict how ligands from one cell population influence gene expression in another [12]. ICELLNET employs a unique strategy of integrating protein–protein interaction data and signaling pathway information to reconstruct cell–cell interaction (CCI) networks, emphasizing intracellular cascades triggered by intercellular communication [13]. Together, these tools represent a rich and diverse methodological landscape, each using statistical, integrative, or knowledge-based strategies to decode complex cell–cell signaling patterns from scRNA-seq data [1,9,14]. Despite their sophistication, current methods often struggle to detect subtle yet biologically important interactions. This limitation is particularly critical in cancer research using tumor-derived scRNA-seq data, in which small differences in gene expression may be overlooked by methods that favor strong or global statistical trends.

Deep learning offers a powerful solution to these problems. For instance, DeepCCI is a recently developed deep-learning-based tool for predicting CCIs from scRNA-seq data. It combines an autoencoder–graph convolutional network-based cell clustering module with an interaction prediction module that integrates residual neural networks and graph convolutional networks [15]. However, DeepCCI lacks explainability in terms of model interpretability, and its dependence on differentially expressed genes may introduce statistical bias, limiting its ability to selectively prioritize interactions specific to certain cell types.

To overcome these limitations, we developed PriorCCI, a deep learning framework designed to prioritize CCIs between specific pairs of cell types. The purpose of this study is to establish a scalable and interpretable deep learning approach that systematically identifies and prioritizes key ligand–receptor interactions between cancer cells and other cell types within the tumor microenvironment (TME) using scRNA-seq data. PriorCCI uses a CNN to integrate gene expression patterns from two cell-type groups and employs Grad-CAM++ for visual interpretation [16] to quantify the contribution of each ligand–receptor pair for the groups based on the learning result by CNN. We applied this method to two independent datasets of lung and colorectal cancers sourced from the Cancer Cell Atlas (CCA) [17,18]. Our analysis identified ligand–receptor pairs uniquely enriched in interactions between tumors and endothelial cells, which may serve as potential targets for angiogenesis-related studies. This approach enables the systematic, cell-type-specific prioritization of CCIs and provides a scalable framework with broad applications in both basic research and translational medicine.

## 2. Results

### 2.1. Input Data Preparation for PriorCCI and Presentation

To investigate the CCIs within the TME, we first constructed two-channel input samples suitable for deep learning analysis. For each pair of cell types (A and B), we randomly selected 100 representative cells and organized their gene expression values into two expression matrices of size 100 × 6296. The 6296 genes corresponded to 3148 ligand–receptor pairs, with group A sorted by ligand–receptor order and group B sorted by reverse (receptor–ligand) order. This mirrored structure allows the resulting data to be interpreted as two-channel image-like inputs with shapes of (100, 6296, 2), preserving the directional nature of the interactions. For each of the 21 pairwise cell type combinations (classes), 1000 samples were generated through random sampling, resulting in 21,000 input samples used to train the PriorCCI model (Figure 1).

To determine class-specific gene importance, we applied the Grad-CAM++ algorithm to the trained CNN model. For each class, the top 25% of the ligand–receptor pairs were selected based on the normalized contribution scores. To evaluate the reproducibility of the model outputs, training was repeated 10 times independently, and the similarity between Grad-CAM++ importance scores across models was assessed using cosine similarity and Spearman correlation.

This analysis was performed using scRNA-seq datasets from two epithelial-origin cancers, non-small cell lung cancer (NSCLC) and colorectal cancer (CRC). Both datasets included cancerous and adjacent normal tissues curated in an integrated atlas format. After preprocessing, 36,601 genes were analyzed in 482,351 NSCLC and 702,657 CRC cells. As described in the Materials and Methods section, cell types were annotated and visualized using UMAP, showing a clear separation among seven major groups: tumor, endothelial, fibroblast, myeloid, T/NK, B, and epithelial cells. All possible pairwise combinations (_7_C_2_ = 21) were considered in the analysis (Figure 2).

To reduce the computational burden and correct for cell type imbalance, we employed geometric sketching using endothelial cell counts (the lowest among cell types: NSCLC, 9990; CRC, 23,742) as the baseline. This resulted in balanced subsets consisting of 104,109 cells for NSCLC and 247,011 for CRC (Table 1). Representative cells from each type were evenly sampled to construct the input data for all pairwise interaction classes.

### 2.2. Performance Evaluation of the CNN Model

To quantitatively evaluate the classification performance, CNN models were trained for 21 cell-pair classes using the generated two-channel inputs. For each of the 10 training runs (initialized with different random seeds), the data were split into training, validation, and test sets in a 64:16:20 ratio. The model performance was assessed using accuracy, precision, recall, F1 score, and area under the ROC curve (AUC).

As shown in Figure 3, both NSCLC and CRC datasets demonstrated steady decreases in training and validation loss and a gradual increase in validation accuracy, reaching ≥ 0.95 and indicating stable convergence without overfitting. These results demonstrate that the model consistently converges during training and achieves strong generalization performance.

As summarized in Table 2, the model accuracy for NSCLC ranged from 0.986 to 0.997, with other performance metrics (precision, recall, and F1 score) exceeding 0.98. For CRC, the accuracy ranged from 0.929 to 0.981, with other metrics exceeding 0.94. Both datasets exhibited low variability across runs and consistently strong performance. The average loss values were also low (0.0289 for NSCLC and 0.154 for CRC), highlighting the robustness and generalization capability of the PriorCCI framework. Notably, the macro-averaged AUC across all classes remained near 1.0 (range: 0.999–1.0), confirming the model’s excellent discriminative power.

The confusion matrices (Figure 3) showed high prediction accuracy across all classes. The ROC curves revealed a strong separation between classes, and standard deviations of the AUC across 10 runs remained below 0.02, further supporting the reproducibility of model performance. These results indicate that PriorCCI effectively learns CCI patterns from single-cell expression data and is capable of reliably distinguishing cell-type pairs within a complex TME.

### 2.3. Model Consistency Across Training Runs

To evaluate the reproducibility and reliability of PriorCCI, Grad-CAM++ importance results were compared across 10 independently trained models. Figure 4 shows the similarity metrics (cosine and Spearman correlations) calculated for the prioritized gene pairs of each model. The mean and standard deviation values for these similarities, excluding self-comparisons, are summarized in Table 3.

Strong consistency in gene rankings was observed among high-interaction cell type pairs, such as Tumor–Endothelial and Tumor–Myeloid, suggesting that PriorCCI steadily captures biologically meaningful interaction patterns. Conversely, a combination of Tumor–Epithelial, especially in CRC, whose cells are from the same tissue origin, showed lower inter-model similarities among 10 models, as evidenced by the reduced cosine and Spearman correlation values shown in Figure 4 and Table 3. This might be the result of the cell-type pair exhibiting weaker or less distinguishable interaction signatures, likely due to the biological similarity between cell types. These findings highlight the model’s ability to reflect biological variability and to distinguish context-specific interactions across different cancer types adaptively.

### 2.4. Functional Validation of Prioritized Gene Pairs Using GSEA

To validate the biological relevance of the genes prioritized by the PriorCCI model, gene set enrichment analysis (GSEA) was performed using the corresponding gene sets [19]. In the NSCLC dataset, interactions between Tumor–Endothelial and Tumor–Myeloid, which are known to play major roles in the TME, were analyzed. In the CRC dataset, Tumor–Fibroblast and Tumor–Myeloid interactions were examined. The enrichment results for these interactions are illustrated in Figure 5, which presents the associated biological pathways identified through GSEA.

In NSCLC, the Tumor–Endothelial interaction showed significant enrichment of angiogenesis-related pathways, including ‘Angiogenesis’, ‘Vascular endothelial cell response’, and ‘Blood vessel morphogenesis’. These results demonstrate that the gene pairs prioritized by PriorCCI are closely related to angiogenesis-associated biological processes. In the Tumor–Myeloid interaction, enrichment was observed in immune-related pathways such as ‘IL-6/JAK/STAT3 Signaling’, ‘Cytokine–cytokine receptor interaction’, ‘Inflammatory Response’, and ‘Chemokine binding’. This indicates that the prioritized gene pairs are involved in the regulation of tumor immune responses.

In CRC, Tumor–Fibroblast interactions showed enrichment in ‘Epithelial Mesenchymal Transition (EMT)’, ‘ECM–receptor interaction’, ‘Collagen receptor activity’, and ‘Extracellular matrix organization’, all of which are representative of cancer-associated fibroblast activation pathways. Tumor–Myeloid interactions were enriched in pathways such as ‘Neutrophil extracellular trap formation’, ‘Macrophage activation’, and ‘Positive regulation of myeloid leukocyte-mediated immunity’.

These findings confirm that the interaction gene pairs derived from PriorCCI reflect key biological characteristics of the TME, rather than being the result of mere mathematical optimization.

### 2.5. Comparison with Existing CCI Analysis Tools on Gene Priorities

To validate the gene pair prioritization, PriorCCI was compared with five established CCI inference tools: CellPhoneDB, ICELLNET, CellChat, NicheNet, and DeepCCI. While most tools rely on statistical or expression-based inferences, DeepCCI uses a deep-learning-based approach. Comparisons focused on the Tumor–Endothelial cell pair in NSCLC and CRC, where PriorCCI showed strong intra-model consistency. The top 25% of gene pairs ranked by Grad-CAM++ (averaged over 10 model runs) intersected with the predictions from each tool. Gene pairs were further filtered using expressing cell fraction (ECF); only gene pairs in which each gene showed the maximal ECF within the relevant cell type were retained. In NSCLC, this process identified 83 validated interacting gene pairs, many of which were not predicted using existing methods.

Figure 6 shows the overlap between PriorCCI and the other tools. Gene pairs on the *x*-axis are ranked by their Grad-CAM++ importance scores, whereas the *y*-axis lists each tool. Tools, such as CellPhoneDB and ICELLNET, detected more overlapping pairs, whereas CellChat and NicheNet detected fewer. Notably, PriorCCI’s top-ranked pairs were concentrated among biologically meaningful interactions, whereas traditional tools yielded a broader, less specific distribution.

### 2.6. Single-Cell Expression of Tumor-Endothelial Gene Pairs

To further validate the specificity of the prioritized gene pairs, ECF heat maps were generated for the top 30 gene pairs in the Tumor–Endothelial CCI (Figure 7a). This analysis highlighted gene pairs with highly distinct expression patterns between tumor and endothelial cells. For comparison, ECF distributions were plotted for the top 10 gene pairs from CellPhoneDB, ICELLNET, CellChat, NicheNet, and DeepCCI.

Notably, some gene pairs frequently prioritized by existing tools, such as APP–CD74, exhibited uniformly high expression across normal immune cell types, suggesting poor specificity for Tumor–Endothelial interactions (Table 4). This underscores the limitations of the previous methods for identifying context-specific CCI.

Figure 7b presents the aligned heat maps of the final prioritized gene pairs, showing the expression specificity for tumor and endothelial cells. Genes strongly expressed in tumor cells were predominantly oncogenes, whereas endothelial-expressed genes were enriched in angiogenesis-related pathways, demonstrating biological plausibility.

Specifically, ITGB3–VWF and ITGAV–VWF were consistently identified as high-priority pairs in the NSCLC dataset, exhibiting strong Grad-CAM++ importance and high ECF values. These gene pairs participate in angiogenic signaling and contribute to tumor growth within the TME [20].

## 3. Discussion

In this study, we propose PriorCCI, a deep-learning-based framework designed to prioritize cell-type-specific CCIs. The framework combines a CNN with Grad-CAM++, enabling the detection of subtle differences in gene expression between cell types, which are often overlooked by conventional statistical methods. We validated PriorCCI using datasets from patients with both NSCLC and CRC. Despite distinct differences in cellular composition and gene expression between these tumor types, PriorCCI demonstrated consistently high classification performance, achieving an average macro-AUC of ≥0.999. These results indicated the strong generalizability of the model across diverse TMEs.

In particular, key CCIs identified in NSCLC, especially the interactions between the tumor and endothelial cells, are strongly associated with angiogenesis-related signaling pathways. These ligand–receptor pairs may serve as promising candidates for the therapeutic targeting of tumor progression. Accordingly, PriorCCI not only enables precise prioritization of intercellular interactions but also facilitates biological interpretation and informs therapeutic strategy development.

Recent lung cancer studies have highlighted the heterogeneity of vascular subtypes within the TME, with growing interest in the role of tumor endothelial cells (TECs) [21]. TECs actively participate in tumor vascular remodeling, suppress immune cell infiltration, and impede drug delivery, thereby contributing to immune evasion and therapeutic resistance [22]. Given these roles, interactions between tumor cells and TECs are considered high-priority targets for therapeutic interventions.

A key strength of PriorCCI is its ability to provide explanations. By incorporating Grad-CAM++, our framework provides intuitive visual interpretations of why certain gene pairs are prioritized, thus improving transparency in deep-learning-based inference. The reproducibility of the top-ranked gene pairs across multiple runs further supported the robustness of the model. To enhance biological credibility, we applied ECF filtering to ensure that the prioritized genes were broadly expressed within the relevant cell populations. While conventional tools often rank gene pairs such as APP–CD74 highly, these genes are broadly expressed across tumor, endothelial, and immune cells [23], making them suboptimal targets. PriorCCI mitigates nonspecific predictions by integrating Grad-CAM++ prioritization with ECF-based filtering.

Although PriorCCI was validated using only lung and colon cancer datasets, its framework is applicable to a broad range of tissue and disease contexts, provided that appropriate scRNA-seq input formats are available. Its potential applications include autoimmune diseases, infectious diseases, and developmental biology.

However, this study has several limitations. First, the model relies solely on transcriptomic data; therefore, protein-level validation is necessary to confirm the functional relevance of the prioritized gene pairs. Second, rare cell populations present in very small numbers pose statistical challenges owing to insufficient sample sizes, which limits interaction inference. This is a common constraint in conventional tools. Future studies should explore strategies to enhance the analysis of rare cell types while maintaining their biological validity. Third, the current CNN architecture uses simple ligand–receptor ordering to structure its input matrix. For example, incorporating functional similarity into matrix organization via interaction networks or clustering may improve CNN learning efficiency and enhance the precision of CCI detection.

In conclusion, PriorCCI is a robust and interpretable framework for prioritizing and interpreting CCIs in scRNA-seq data. Given its interpretability and scalability, PriorCCI has potential for integration into translational pipelines aimed at therapeutic target discovery, such as for immune-oncology or anti-angiogenic drug development. Its reliable performance and broad utility make it a valuable tool for cancer research as well as for exploring complex physiological and pathological processes.

## 4. Materials and Methods

### 4.1. Data Preprocessing and Sampling of Representative Cells for Each Cell Type

Datasets for NSCLC and CRC were obtained from CCA and developed through our prior collaborative research. Cell-type annotations were conducted using a scanpy-based preprocessing pipeline [24] in combination with manual curation of marker genes following automatic annotation by SingleR (v.1.4.0) [25] and CellTypist (v.1.6.3) [26].

Tumor cells were identified based on copy number variation (CNV) scores calculated using InferCNVpy (v.0.4.2, https://github.com/icbi-lab/infercnvpy, accessed on 1 August 2023) [27]. Cells with a CNV z-score ≥ 1 compared to normal cells within each cluster were classified as tumor cells.

As illustrated in Figure 1, once cell types were pre-annotated in the scRNA-seq data, subsampling was performed using geometric sketching [28] to reduce the computational burden. This approach utilizes the coordinate positions and metadata from each cell-type cluster. To construct the input data for CCI classification, we extracted 100 representative cells per cluster based on gene expression and organized their profiles by ligand–receptor pairs. For each pairwise combination of cell type clusters, we generated expression matrices (100 cells per cluster) and performed 1000 random samplings per combination to comprehensively capture the overall co-expression patterns.

### 4.2. CNN in PriorCCI

To construct a comprehensive list of biologically relevant ligands and receptors, we integrated and curated data from CellPhoneDB and ICELLNET, resulting in 3148 ligand–receptor pairs composed of 755 receptor and 893 ligand genes.

As shown in Figure 1, the form of input for multiclass classification using a CNN is as follows: The final matrix consisted of 6296 pairs, including 3148 ligand–receptor pairs for the two cell types and 3148 pairs in the opposite direction, forming a matrix of 100 cells each (100, 6296, 2). The CNN comprises four convolution steps, with the initial three steps serving to reduce the expression values of the cells to a single value. The first convolution layer was performed for each ligand–receptor pair from the two cell types. The second convolution layer reduced the values of the 10 cells to a single value. The third convolution layer repeated this process. The filters applied in each convolution were 8, 16, 16, and 32 filters. The kernel filters move in strides (1,1), (10,1), (10,1), and (1,4), respectively.

For a standard 2D convolution operation, let the input feature map be noted as $X\in R^{H\times W\times C}$, where *H*, *W*, and *C* represent the height, width, and number of input channels, respectively. A set of learnable convolutional kernels (filters) is noted as $W\in R^{K_{H}\times K_{W}\times C\times C^{\prime }}$, where *K_H_* and *K_W_* are the kernel height and width, and C′ is the number of output channels. Each output feature map *Y*^(*k*)^ is obtained by computing the weighted sum of the local region of the input, followed by the addition of a learnable bias term *b*^(*k*)^, *k*-th output channel. This is expressed as follows:(1)$$Y_{i,j}^{\left(k\right)}=\sum_{m=0}^{K_{H}\;-\;1}\sum_{n=0}^{K_{W}\;-\;1}\sum_{c=0}^{C\;-\;1}X_{i+m,j+n}^{\left(c\right)}\cdot W_{m,n}^{\left(c,k\right)}+b^{\left(k\right)}$$

Here, *i* and *j* indexes the spatial position of the output, and *k*∈[1,*C*′] indexes the output channel.

In all four convolutions, to mitigate the gradient vanishing problem, the activation function used to impart nonlinearity was a Rectified Linear Unit (ReLU) that passed negative numbers as 0 and positive numbers unchanged [29].

MaxPooling was performed before and after the fourth convolution in order to further compress the convolution results, reduce the spatial dimensionality, and ensure that only important features remained. In addition, a batch normalization process was incorporated to normalize the mean and variance on a mini-batch basis to mitigate the problems of gradient runaway or vanishing. GlobalAveragePooling2D calculates the mean value of the entire feature map for each channel, thereby entirely removing spatial information and leaving only the vector information to be sent to the classifier. This was followed by a dense ReLU and two dropout processes that randomly removed 40% of the neurons to prevent overfitting. To obtain class probabilities, the final layer applies a softmax activation function using the following equation:(2)$$\;Softmax\left(z_{i}\right)=\frac{e^{z_{i}}}{\sum_{j=i}^{K}e^{z_{j}}},\;\;\;\;for\;i=1,2,\ldots ,K$$

Here, *z_i_* is the logit corresponding to the *i*-th class, and *K* is the total number of classes. The softmax function converts the raw outputs into a probability distribution across classes, ensuring that the outputs sum up to one.

The model uses Sparse Categorical Cross-Entropy as the loss function for training multiclass classification. The loss of a single sample is defined as follows:(3)$$\;L_{CE}=-\log\left({\hat{y}}_{c}\right)$$
where ${\hat{y}}_{c}$ denotes the predicted probability of class *c* obtained from the softmax output layer.(4)$$\;{\hat{y}}_{i}=\frac{e^{z_{i}}}{\sum_{j=1}^{C}e^{z_{j}}},\;\;\;\;for\;i=1,2,\ldots ,C$$
where *z_i_* is the logit of the pre-softmax activation for class *i* and *C* is the total number of classes.

Optimization was performed using the Adam optimizer [30], with learning rate decay applied via plateau detection. It adapts the learning rate for each parameter based on the first and second moments of the gradient. The update rule for parameter *θ* at time step *t* is:(5)$$m_{t}=\beta_{1}m_{t-1}+\left(1-\beta_{1}\right)\nabla_{\theta }L_{t}$$(6)$$\upsilon_{t}=\beta_{2}\upsilon_{t-1}+\left(1-\beta_{2}\right){\left(\nabla_{\theta }L_{t}\right)}^{2}$$(7)$${\hat{m}}_{t}=\frac{m_{t}}{1-\beta_{1}^{t}},\;\;\;\;{\hat{\upsilon }}_{t}=\frac{\upsilon_{t}}{1-\beta_{2}^{t}}$$(8)$$\theta_{t}=\theta_{t-1}-\eta \cdot \frac{{\hat{m}}_{t}}{\sqrt{{\hat{\upsilon }}_{t}}+\epsilon }$$
where *η* is the learning rate (set to 1 × 10^−4^ in our experiments), and the default hyperparameters are set as follows: *β*_1_ = 0.9, *β*_2_ = 0.9999, and *ϵ* = 10^−8^.

The models were evaluated using the accuracy, precision, recall, F1 score, and area under the ROC curve (AUC) as the primary metrics. In multiple classifications, the AUC was computed using the one-vs.-rest strategy and macro-averaged across all classes. In addition, to ensure robustness, the stability and consistency of the 10 models were evaluated through 10 learning repetitions using random seeds.

### 4.3. Similarity Calculation Within Models

To compare gene importance across different models, we computed cosine similarity:(9)$$cosine\_similarity\left(A,B\right)=\frac{\sum_{i=1}^{n}A_{i}B_{i}}{\sqrt{\sum_{i=1}^{n}A_{i}^{2}}\sqrt{\sum_{i=1}^{n}B_{i}^{2}}}$$

We also used Spearman rank correlation to assess monotonic relationships:(10)$$\rho =1-\frac{6\sum d_{i}^{2}}{n\left(n^{2}-1\right)}$$
where *d_i_* is the rank difference between corresponding elements.

### 4.4. Visual Interpretation with Grad-CAM++ in PriorCCI

Following CNN learning in PriorCCI, this is the most central method for visually interpreting the priorities of gene pairs. This can be summarized in six steps to determine the priority of major gene combinations:Model and class definition: The first step was to define the model and class. Let *f*:*X*→*R*^*C*^ be the trained CNN model, where *X*∈ℝ^*H*×*W*×*D*^ is the input (e.g., ligand–receptor pixel image), *C* is the number of output classes. We denote the output logit (before softmax) for class *c* as *y*^*c*^ = *f*_*c*_(*X*).Grad-CAM++ computation: In the second step, an importance map calculation based on Grad-CAM++ is performed. Let *A*^*k*^∈ℝ^*H*′×*W*′^ be the *k*-th feature map at the last convolutional layer. The importance weight $\alpha_{k}^{c}$ for class *c* is computed via Grad-CAM++ as:(11)$$\alpha_{k}^{c}=\sum_{i,j}\frac{\partial^{2}y^{c}}{{\left(\partial A_{ij}^{k}\right)}^{2}}\cdot ReLU\left(\frac{\partial y^{c}}{\partial A_{ij}^{k}}\right)$$

Then, the Grad-CAM++ heatmap $L_{\mathrm{Grad}-\mathrm{CAM}++}^{c}$ is:(12)$$L_{\mathrm{Grad}-\mathrm{CAM}++}^{c}=\backslash ReLU\left(\sum_{k}\alpha_{k}^{c}A^{k}\right)$$

This heatmap is normalized for visualization:(13)$$\tilde{L^{c}}=\frac{L^{c}-\min\left(L^{c}\right)}{\max\left(L^{c}\right)-\min\left(L^{c}\right)}$$

3.Classwise average of CAMs: The third step is the calculation of the class-specific average of the CAM. Given *N* samples from class *c*, the classwise mean of the CAM is:


(14)
$$\bar{L^{c}}=\frac{1}{N}\sum_{n=1}^{N}\tilde{L_{n}^{c}}$$


4.Extraction of ligand–receptor importance: In the fourth step, the importance of each ligand–receptor pair must be extracted. Given the predefined ligand–receptor index ${\left\{\left(l_{i},r_{i}\right)\right\}}_{i=1}^{G}$, the CAM weight for pair *i* is:



(15)
$$w_{i}^{c}=\bar{L^{c}}\left[l_{i},r_{i}\right]$$



5.Statistical analysis: The fifth step was the statistical analysis of gene pairs with the top 5% importance values. Let $w_{i}^{\left(j\right)}$ be the weight of the gene pair *i* in model run *j* (total *M* runs). Filtering the top 5% per model, we define(16)$$\mu_{i}=\frac{1}{M_{i}}\sum_{j=1}^{M_{i}}w_{i}^{\left(j\right)},\;\;\;\;\sigma_{i}^{2}=\frac{1}{M_{i}}\sum_{j=1}^{M_{i}}{\left(w_{i}^{\left(j\right)}-\mu_{i}\right)}^{2}$$
where *M*_*i*_ is the number of models where pair *i* is in the top 5%. And then we define the coefficient of variation (CV) and median values.6.Final ranking: The final step in the process entails the acquisition and organization of the information set, denoted by $\left(\mu_{i},\sigma_{i}^{2},{\mathrm{CV}}_{i},{\mathrm{Med}}_{i},M_{i}\right)$, and its subsequent arrangement in descending order of *M_i_* or *μ_i_*.

### 4.5. Gene Filtering with ECF

While Grad-CAM++ highlights the gene pair importance between cell types, further validation of the gene expression patterns is necessary. In scRNA-seq data, it has been noted that the proportion of cells expressing a gene is often more informative than averaged expression levels [18]. Therefore, for all ligand–receptor pairs prioritized by Grad-CAM++, we assessed the ECF in the relevant cell type. Only gene pairs with ECF values in the upper quantiles (top 25%) of the distribution were retained, allowing for a more biologically meaningful interpretation of key interactions.

## Figures and Tables

**Figure 1 ijms-26-07110-f001:**
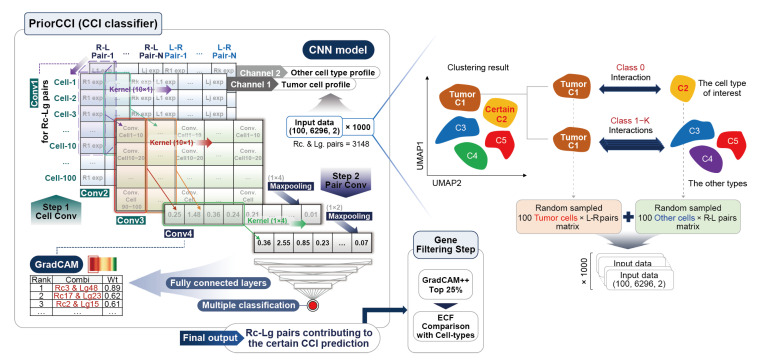
Overview of the PriorCCI framework. Each interaction class is defined by specific cell type combinations identified via scRNA-seq data clustering. Two-channel expression matrices are constructed from ligand–receptor gene pairs across sampled cell profiles. The CNN model applies two steps of convolutions and Grad-CAM++ to prioritize biologically relevant ligand–receptor interactions, followed by gene filtering.

**Figure 2 ijms-26-07110-f002:**
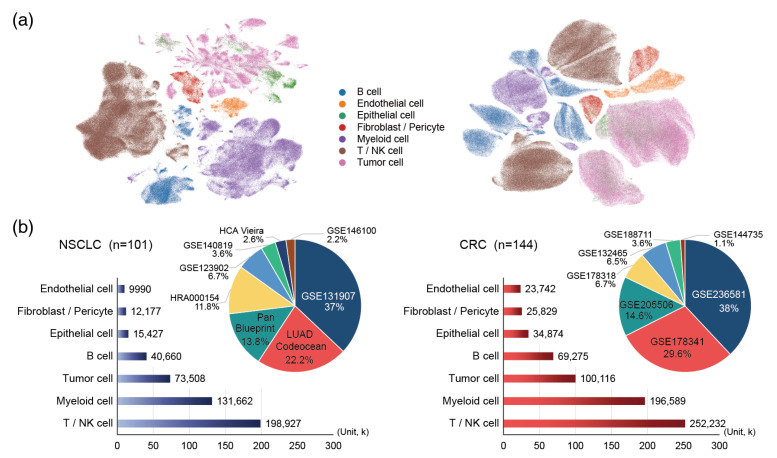
Summary of all datasets used for PriorCCI (**a**) UMAP presentation with clusters representing the major cell types in the NSCLC and CRC datasets, respectively: NSCLC, Non-small cell lung cancer; CRC, Colorectal cancer. (**b**) Bar and pie plots for showing the composition of cell-type-specific cell counts and data-source-wise proportions, respectively. NSCLC is on the left, and CRC is on the right.

**Figure 3 ijms-26-07110-f003:**
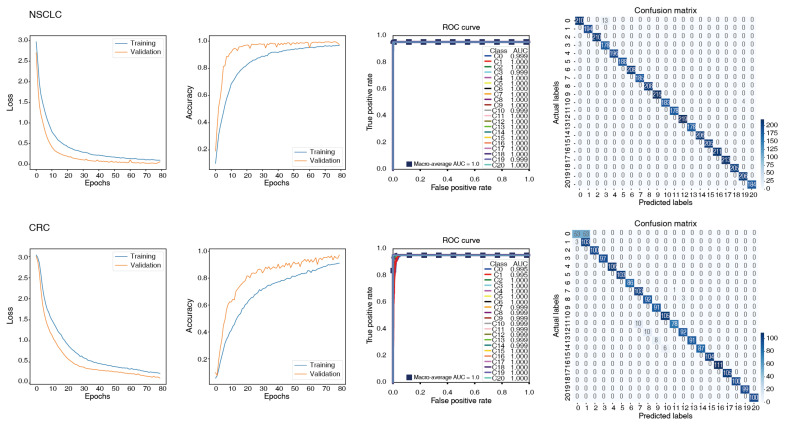
Performance of PriorCCI. The learning curves for loss and accuracy according to epochs, using the training and validation sets, are shown on the left. The right side displays ROC curves and the associated macro-average AUC, as well as the confusion matrix. The upper plots are for NSCLC, and the lower plots are for CRC.

**Figure 4 ijms-26-07110-f004:**
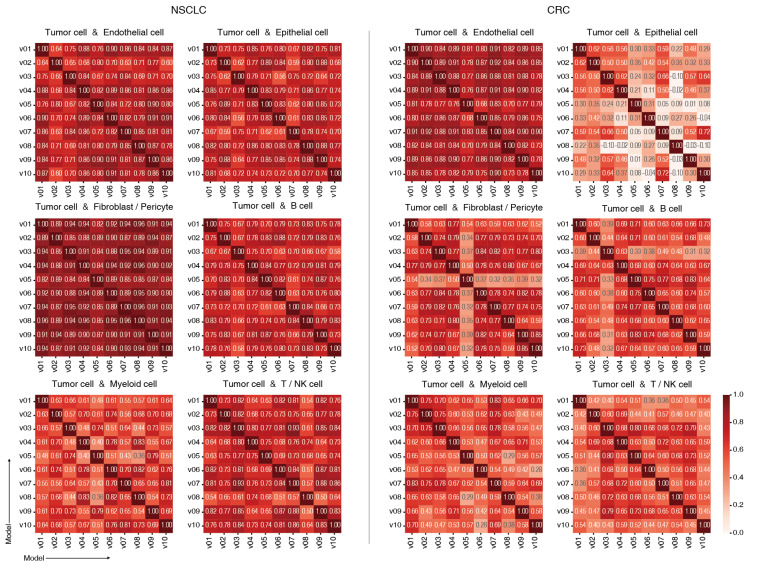
Similarity across 10 models. Heatmaps show the similarity of ligand–receptor importance values across models for each cell-type combination involving tumor cells. Similarity was computed using both cosine and Spearman methods; the results shown here are based on Spearman correlation.

**Figure 5 ijms-26-07110-f005:**
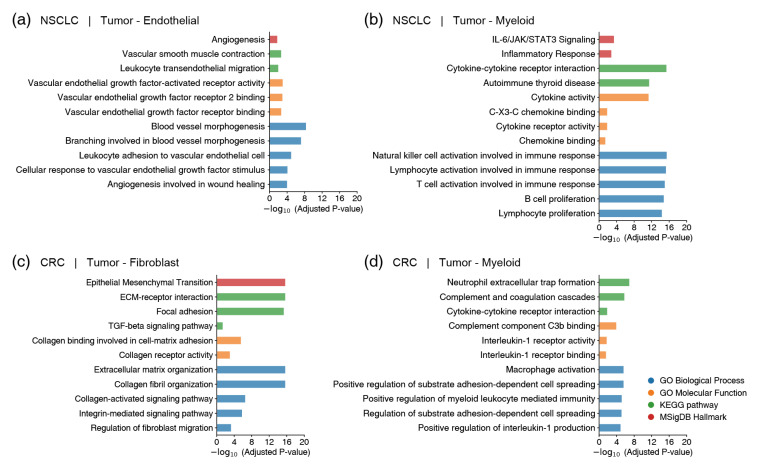
Functional validation of PriorCCI-prioritized gene pairs through enrichment analysis. (**a**) Bar plot showing significantly enriched terms associated with gene pairs from Tumor–Endothelial interactions in NSCLC. (**b**) Enriched pathways for Tumor–Myeloid interactions in NSCLC. (**c**) Functional terms for Tumor–Fibroblast interactions in CRC. (**d**) Immune-related enrichment in Tumor–Myeloid interactions in CRC. Each bar indicates an enriched term from Gene Ontology (GO Biological Process and GO Molecular Function), KEGG pathway, or MSigDB Hallmark. Color legend corresponds to the source of each term.

**Figure 6 ijms-26-07110-f006:**
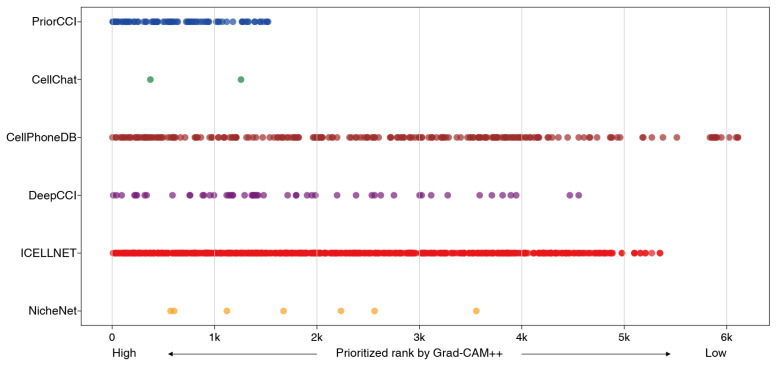
Distribution of gene-pair rankings assigned by each method relative to prioritization in PriorCCI. The horizontal strip plot compares the ranks of ligand–receptor pairs predicted by each CCI analysis method for the Tumor–Endothelial cell interaction. Each dot represents a gene pair detected by one of the tools. The *x*-axis indicates the Grad-CAM++-based rank in PriorCCI, with higher-priority pairs positioned on the left (top 25%).

**Figure 7 ijms-26-07110-f007:**
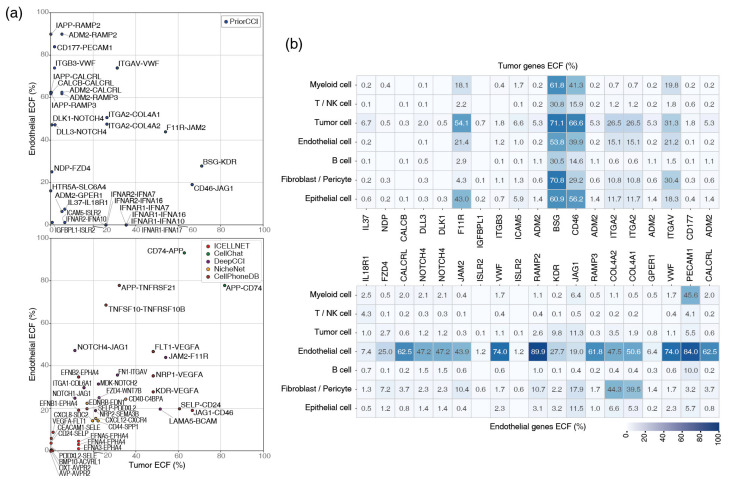
ECF status of ligand–receptor pair candidates specific for the Tumor–Endothelial cell interaction (**a**) Candidates of gene–gene pairs from PriorCCI (upper) and other methods (lower) are shown as ECF (%) with respect to endothelial cells (*y*-axis) and tumor cells (*x*-axis). (**b**) The ECF values of the PriorCCI candidates according to cell type. The genes from tumor cells (top) and endothelial cells (bottom) are compared in other cell types after filtering out genes with an ECF under 0.1% in all cell types.

**Table 1 ijms-26-07110-t001:** Adjusted cell counts for each cell type after geometric sketching for subsampling.

Cell Type	NSCLC		CRC	
	Not Applied	Applied	Not Applied	Applied
T/NK	198,927	18,594	252,232	43,586
Tumor	131,662	17,856	196,589	42,991
B	73,508	17,729	100,116	42,142
Myeloid	40,660	15,502	69,275	36,787
Epithelial	15,427	12,734	34,874	32,111
Fibroblast/Pericyte	12,177	11,704	25,829	25,652
Endothelial	9990	9990	23,742	23,742
Total	482,351	104,109	702,657	27,001

T/NK: T/Natural Killer cells.

**Table 2 ijms-26-07110-t002:** Performance of 10 CNN models for each cancer type.

Model	NSCLC						CRC					
	Loss	Accuracy	Precision	Recall	F1 Score	Macro AUC	Loss	Accuracy	Precision	Recall	F1 Score	Macro AUC
v1	0.021	0.993	0.994	0.993	0.993	1.000	0.141	0.954	0.961	0.954	0.953	0.999
v2	0.043	0.983	0.985	0.983	0.983	1.000	0.200	0.942	0.945	0.942	0.942	0.999
v3	0.018	0.997	0.997	0.997	0.997	0.999	0.075	0.977	0.979	0.977	0.977	0.999
v4	0.021	0.994	0.994	0.994	0.994	1.000	0.198	0.949	0.952	0.949	0.948	0.999
v5	0.029	0.989	0.990	0.989	0.989	1.000	0.177	0.943	0.945	0.943	0.943	0.999
v6	0.043	0.986	0.987	0.986	0.986	1.000	0.191	0.944	0.946	0.944	0.944	0.999
v7	0.027	0.991	0.991	0.991	0.991	0.999	0.157	0.949	0.950	0.949	0.949	0.999
v8	0.032	0.988	0.989	0.988	0.988	1.000	0.171	0.929	0.942	0.929	0.927	0.999
v9	0.026	0.992	0.992	0.992	0.992	0.999	0.168	0.955	0.957	0.955	0.955	0.999
v10	0.028	0.992	0.993	0.992	0.992	1.000	0.064	0.981	0.981	0.981	0.980	1.000
Avg.	0.029	0.991	0.991	0.991	0.991	1.000	0.154	0.952	0.956	0.952	0.952	0.999

**Table 3 ijms-26-07110-t003:** Summary of model-to-model similarities across CCI classes based on the importance scores by Grad-CAM++.

Class No.	NSCLC				CRC			
	Cosine		Spearman		Cosine		Spearman	
	Mean	SD	Mean	SD	Mean	SD	Mean	SD
0	0.867	0.049	0.652	0.090	0.884	0.047	0.632	0.136
1	0.902	0.030	0.672	0.100	0.854	0.035	0.603	0.113
2	0.899	0.058	0.830	0.062	0.904	0.027	0.822	0.043
3	0.970	0.011	0.902	0.038	0.907	0.022	0.731	0.088
4	0.883	0.042	0.700	0.104	0.930	0.025	0.848	0.060
5	0.934	0.024	0.756	0.080	0.841	0.053	0.474	0.172
6	0.895	0.044	0.867	0.039	0.835	0.109	0.628	0.149
7	0.959	0.016	0.944	0.019	0.766	0.086	0.321	0.223
8	0.943	0.022	0.915	0.035	0.845	0.057	0.658	0.106
9	0.957	0.014	0.904	0.036	0.940	0.015	0.914	0.030
10	0.717	0.113	0.418	0.140	0.861	0.064	0.734	0.074
11	0.822	0.077	0.671	0.098	0.892	0.047	0.804	0.056
12	0.894	0.039	0.624	0.114	0.857	0.036	0.662	0.100
13	0.819	0.058	0.557	0.130	0.825	0.086	0.657	0.157
14	0.861	0.035	0.655	0.068	0.902	0.026	0.757	0.085
15	0.883	0.044	0.662	0.093	0.819	0.085	0.565	0.172
16	0.958	0.011	0.884	0.047	0.811	0.049	0.668	0.100
17	0.779	0.095	0.580	0.099	0.845	0.061	0.606	0.120
18	0.924	0.025	0.737	0.098	0.919	0.026	0.834	0.058
19	0.922	0.022	0.751	0.069	0.824	0.072	0.587	0.116
20	0.867	0.049	0.652	0.090	0.818	0.122	0.561	0.117

**Table 4 ijms-26-07110-t004:** Cell type-specific ECF (%) of gene candidates from other CCI tool results.

Cell Type	Gene Candidates					
	APP	CD74	FLT1	TNFRSF21	TNFSF10	TNFRSF10B
T/NK	29.4	95.9	7.9	10.4	24.1	14.2
Tumor	3.6	74.1	0.4	0.3	12.2	3.1
B	62.8	81.8	1.1	32.2	39.9	25.9
Myeloid	77.8	93.1	46.6	2.7	68.6	13.7
Epithelial	6.9	99.1	0.5	0.7	9.1	3.8
Fibroblast/Pericyte	57.7	60.0	1.2	12.7	20.2	9.7
Endothelial	45.6	87.5	0.7	13.9	30.8	17.4

T/NK: T/Natural Killer cells.

## Data Availability

The NSCLC and CRC datasets were obtained from CCA [17,18] and are available on ZENODO at https://zenodo.org/records/10651059. The source code for PriorCCI has been uploaded to our Github site, https://github.com/nccpai/PriorCCI.

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
