# Peer review of "PriorCCI: Interpretable Deep Learning Framework for Identifying Key Ligand–Receptor Interactions Between Specific Cell Types from Single-Cell Transcriptomes"

_ijms, 2025, doi:10.3390/ijms26157110_

Round 1

Reviewer 1 Report

Comments and Suggestions for Authors

The authors applied DL to construct prediction for prioritizing cell- 2cell interactions in single-cell transcriptomes. This is an interesting study. However, there are some issues should be addressed before it could be re-considered for publication.

  1. The study only collected dataset of r NSCLC and CRC .  Is this framwork could be applied for other diseases?
  2. More validations are necessary. And a case of this applicance is suggested.
  3. Comparisons with existed study are necessary.
  4. How to use this deep-learning framework? Any online server or software?

Author Response

Comments 1. The study only collected dataset of NSCLC and CRC. Is this framework could be applied for other diseases?

Response 1: Thank you for this important question. Although our current study focused on NSCLC and CRC due to the availability of well-annotated and large-scale single-cell datasets, the PriorCCI framework is generalizable to other diseases. The model accepts any scRNA-seq data formatted with proper cell-type annotations and sufficient ligand–receptor coverage. In fact, the CNN structure and Grad-CAM++ prioritization steps are disease-agnostic. We have clarified this point in the revised manuscript (Discussion section), and emphasized the potential for application to other pathological and physiological contexts such as autoimmune or infectious diseases.

Comments 2. More validations are necessary. And a case of this application is suggested.

Response 2: We thank the reviewer for this insightful suggestion. In response, we have incorporated several validation strategies to enhance the robustness and biological relevance of PriorCCI’s results. These include:

(i) Multi-run reproducibility checks using cosine and Spearman similarity metrics to demonstrate model stability across ten independent runs;

(ii) ECF (Expressing Cell Fraction) filtering to confirm that predicted ligand–receptor pairs exhibit cell-type-specific expression patterns, enhancing biological interpretability;

(iii) Comparative analysis against five established CCI tools (CellPhoneDB, ICELLNET, CellChat, NicheNet, and DeepCCI) to benchmark the distinctiveness and specificity of our predictions; and

(iv) New analyses based on Gene Set Enrichment Analysis (GSEA) performed on the top-ranked gene sets derived from PriorCCI to provide additional evidence of functional relevance. This GSEA validation was newly added in Section 2.4 in the revised manuscript specifically in response to the reviewer’s comment, and it further supports that the prioritized gene sets are significantly enriched in biological pathways related to angiogenesis and tumor progression.

Additionally, we present a focused case study on the Tumor–Endothelial interaction, highlighting angiogenesis-related ligand–receptor pairs such as ITGB3–VWF and ITGAV–VWF, both of which are supported by the literature and align with known mechanisms in the tumor microenvironment. These enhancements are now reflected in the revised Results and Discussion sections.

Comments 3. Comparisons with existing studies are necessary.

Response 3: We appreciate the reviewer's suggestion to clarify how PriorCCI compares to existing methods. While benchmarking studies (e.g., Liu et al. 2022 in Genome Biology and Xie et al. 2023 in Biomolecules) have systematically highlighted the strengths and limitations of statistical/network-based CCI tools, including CellChat, CellPhoneDB, ICELLNET and NicheNet, with respect to metrics such as spatial consistency and computational performance, our objective differs. Rather than conducting a global performance benchmark, our focus is on demonstrating how PriorCCI’s purpose-driven prioritization identifies Receptor-Ligand pairs that are uniquely emphasized by our model but not highlighted by other tools. To this end, in Section 2.5, we compare the top-ranked Receptor-Ligand interactions identified by PriorCCI against those from CellPhoneDB, ICELLNET, CellChat, NicheNet, and DeepCCI. We show that, while the baseline tools may detect a broad set of interactions, PriorCCI consistently highlights biologically relevant, cell-type specific gene pairs, such as ITGB3–VWF in Tumor–Endothelial communication, that are either deprioritized or overlooked by other methods. This targeted comparison confirms that PriorCCI adds interpretability and specificity beyond conventional rankings, aligning with our stated goal and moving beyond benchmarking to practical relevance.

Comments 4. How to use this deep-learning framework? Any online server or software?

Response 4: We thank the reviewer for pointing this out. We have made the code and instructions publicly available on GitHub: https://github.com/nccpai/PriorCCI, as described in “Data Availability Statement” and also referenced in the ‘Discussion’ in our manuscript. The repository includes a sample of input format, model weights, and scripts for running predictions and visualizing results.

Reviewer 2 Report

Comments and Suggestions for Authors

In manuscript Authors developed PriorCCI, a deep-learning framework that leverages a convolutional neural network alongside Grad-CAM++. PriorCCI effectively prioritizes interactions between cancer and other cell types within the tumor microenvironment and accurately identifies biologically significant interactions related to angiogenesis. Manuscript is interesting, corresponds to the journal theme, but the paper has some problems.

Some comments:

-I did not see the purpose of this paper. The Authors should note in the title the subject of their research and expand this into the purpose of the paper, which is not there yet.

-Authors wrote “As shown in Figure 3, both NSCLC and CRC datasets demonstrated steady decreases in training and validation loss and a gradual increase in validation accuracy, reaching ≥ 0.95 and indicating stable convergence without overfitting” Please add the explanation.

- Text “Conversely, combinations such as tumor-epithelial cells, composed of cell types from the same tissue origin, showed lower similarity, especially in CRC, indicating weaker or less distinct interaction signals. These patterns reflect the biological variability and further validate the model’s ability to adapt to different cancer types” not so clear, what confirms these patterns?

- What are the next steps for this research? A brief discussion of future directions could add depth to the conclusion.

-Please indicate the advantage of your research over similar ones.

-I would suggest adding a sentence in article to discuss how this interpretable deep learning framework could be relevant for real-life scenario.

- Figs. – edit the text part of the figures, increase the font size.

Author Response

Comments 1. I did not see the purpose of this paper. The Authors should note in the title the subject of their research and expand this into the purpose of the paper, which is not there yet.

Response 1: We appreciate the reviewer’s constructive feedback regarding the clarity of the paper’s purpose. In response, we have revised the title, “PriorCCI: Interpretable Deep Learning Framework for Identifying Key Ligand–Receptor Interactions between Specific Cell Types from Single-Cell Transcriptomes”, to more explicitly convey the study’s core objective—developing an interpretable deep learning framework for prioritizing ligand–receptor interactions specific to certain cell-type combinations based on single-cell RNA-seq data.

Furthermore, in the Abstract and the final paragraph of the Introduction, we now clearly articulate the purpose of this work:

“to provide a scalable, explainable, and biologically meaningful framework to identify and prioritize key ligand–receptor interactions between defined cell-type pairs using scRNA-seq data, particularly in complex environments such as tumors.”

We hope this revision more effectively communicates the contribution and scope of the study.

Comments 2. Authors wrote “As shown in Figure 3, both NSCLC and CRC datasets demonstrated steady decreases in training and validation loss and a gradual increase in validation accuracy, reaching ≥ 0.95 and indicating stable convergence without overfitting” Please add the explanation.

Response 2: We have now expanded the explanation in the Results section (Section 2.2) to clarify the significance of the decrease in loss and the increase in accuracy. Specifically, we emphasize that validation accuracy reaching ≥0.95 over multiple runs, with low loss and AUC close to 1.0, indicates stable convergence and reliable generalization.

Comments 3. Text “Conversely, combinations such as tumor-epithelial cells, composed of cell types from the same tissue origin, showed lower similarity, especially in CRC, indicating weaker or less distinct interaction signals. These patterns reflect the biological variability and further validate the model’s ability to adapt to different cancer types” not so clear, what confirms these patterns?

Response 3: We appreciate the reviewer’s comment and have revised the sentence in Section 2.3 to improve clarity and better explain the observed patterns. The updated text now reads:

“Conversely, a combination of Tumor–Epithelial cells in CRC, composed of cell types originating from the same tissue, showed lower inter-model similarity across ten independent runs, as evidenced by the reduced cosine and Spearman correlation values shown in Figure 4 and Table 3. This likely reflects that these cell types—due to their shared lineage—exhibit less distinguishable ligand–receptor expression patterns, resulting in weaker or less distinct interaction signals. These findings support the model’s capacity to reflect underlying biological characteristics and adaptively capture the context-specific nuances of cell–cell interactions across different cancer types.”

We believe this revision more clearly addresses the reviewer’s concern by directly linking the observed similarity patterns to the biological nature of the involved cell types and by pointing to the supporting quantitative results in the figures and tables.

Comments 4. What are the next steps for this research? A brief discussion of future directions could add depth to the conclusion.

Response 4: We appreciate the reviewer’s suggestion to provide a forward-looking perspective. Accordingly, we have added a paragraph to the Discussion section that outlines several future directions for this work. These include:

(i) extending the application of PriorCCI to other diseases such as autoimmune and infectious disease,

(ii) incorporating emerging data types such as spatial transcriptomics and proteomics to enhance the contextual resolution of inferred interactions, and

(iii) refining the input structure of our CNN framework using biological priors such as pathway-level relationships or spatial proximity metrics to further improve interpretability and performance.

We believe these directions not only reflect the flexibility of our approach but also indicate its potential for broader utility across diverse biomedical settings.

Comments 5. Please indicate the advantage of your research over similar ones.

Response 5: Thank you for highlighting this important aspect. In the revised manuscript, we have expanded the Discussion section to better articulate the key advantages of PriorCCI compared to existing CCI analysis tools. Unlike conventional approaches that rely predominantly on global statistical associations, PriorCCI was specifically designed to prioritize biologically meaningful receptor–ligand pairs between specific cell-type combinations using an interpretable deep learning framework.

Our primary goal was not to replicate existing benchmarking efforts focused on overall performance metrics, which have already been extensively covered in prior studies (e.g., Liu et al. 2022 in Genome Biology and Xie et al. 2023 in Biomolecules), but rather to demonstrate how the top-ranked gene pairs identified by PriorCCI differ from those of other tools in terms of biological specificity and interpretability. This was achieved by:

(i) incorporating Grad-CAM++ to reveal the gene-level contribution in each prediction,

(ii) evaluating model robustness through inter-run consistency,

(iii) applying ECF filtering to highlight cell-type-specific expression patterns and reduce false positives, and

(iv) uncovering subtle but biologically important gene expression patterns that are often missed by statistical averaging.

Through comparative analysis with five widely used CCI tools (CellPhoneDB, ICELLNET, CellChat, NicheNet, and DeepCCI), we showed that PriorCCI identifies unique, high-confidence interaction pairs—such as ITGB3–VWF and ITGAV–VWF—that are closely associated with known angiogenic pathways. These results reinforce the strength of our framework in both prioritization and biological relevance, which we believe sets PriorCCI apart from traditional methods.

Comments 6. I would suggest adding a sentence in article to discuss how this interpretable deep learning framework could be relevant for real-life scenario.

Response 6: We thank the reviewer for pointing out the importance of connecting our research to real-world applications. To address this, we have added the following statement to the Conclusion section:

“Given its interpretability and scalability, PriorCCI has potential for integration into translational pipelines aimed at therapeutic target discovery, such as for im-mune-oncology or anti-angiogenic drug development. Its reliable performance and broad utility make it a valuable tool for cancer research as well as for exploring com-plex physiological and pathological processes.”

This addition is intended to clarify the potential impact of our framework in therapeutic discovery and biomedical research beyond academic settings.

Comments 7. Figs. – edit the text part of the figures, increase the font size..

Response 7: Thank you for this observation. We have revised “all” figure panels to ensure appropriate resolution and increased font size for improved readability, especially in Figures 1, 3, 4, and 7.

Round 2

Reviewer 1 Report

Comments and Suggestions for Authors

The authors well answered my concerns

Reviewer 2 Report

Comments and Suggestions for Authors

Accept.

Comments on the Quality of English Language

Accept